# Whole Genome Sequence Analysis of *Mycobacterium bovis* Cattle Isolates, Algeria

**DOI:** 10.3390/pathogens10070802

**Published:** 2021-06-24

**Authors:** Fatah Tazerart, Jamal Saad, Naima Sahraoui, Djamel Yala, Abdellatif Niar, Michel Drancourt

**Affiliations:** 1Laboratoire d’Agro Biotechnologie et de Nutrition des Zones Semi Arides, Université Ibn Khaldoun, Tiaret 14000, Algeria; fatah_tazrart@hotmail.com; 2Institut des Sciences Vétérinaires, Université de Blida 1, Blida 09000, Algeria; nasahraoui@gmail.com; 3Institut Hospitalo-Universitaire Méditerranée Infection, 13005 Marseille, France; jsaad270@gmail.com; 4Faculté de Médecine, Aix-Marseille-Université, IHU Méditerranée Infection, 13005 Marseille, France; 5Laboratoire National de Référence pour la Tuberculose et Mycobactéries, Institut Pasteur d’Algérie, Alger 16015, Algeria; djamyala@yahoo.fr; 6Laboratoire de Reproduction des Animaux de la Ferme, Université Ibn Khaldoun, Tiaret 14000, Algeria; ameurh65@gmail.com

**Keywords:** *Mycobacterium tuberculosis*, *Mycobacterium bovis*, tuberculosis, cattle, Algeria

## Abstract

*Mycobacterium bovis* (*M. bovis*), a *Mycobacterium tuberculosis* complex species responsible for tuberculosis in cattle and zoonotic tuberculosis in humans, is present in Algeria. In Algeria however, the *M. bovis* population structure is unknown, limiting understanding of the sources and transmission of bovine tuberculosis. In this study, we identified the whole genome sequence (WGS) of 13 *M. bovis* strains isolated from animals exhibiting lesions compatible with tuberculosis, which were slaughtered and inspected in five slaughterhouses in Algeria. We found that six isolates were grouped together with reference clinical strains of *M. bovis* genotype-Unknown2. One isolate was related to *M. bovis* genotype-Unknown7, one isolate was related to *M. bovis* genotype-Unknown4, three isolates belonged to *M. bovis* genotype-Europe 2 and there was one new clone for two *M. bovis* isolates. Two isolates from Blida exhibited no pairwise differences in single nucleotide polymorphisms. None of these 13 isolates were closely related to four zoonotic *M. bovis* isolates previously characterized in Algeria. In Algeria, the epidemiology of bovine tuberculosis in cattle is partly driven by cross border movements of animals and animal products.

## 1. Introduction

Bovine tuberculosis is a chronic, deadly infection most often caused by *Mycobacterium bovis* (*M. bovis*) and *Mycobacterium caprae* (*M. caprae*), two species belonging to the *Mycobacterium tuberculosis* complex [1,2]. *M. bovis* most often infects cattle but also causes tuberculosis in other animal species and is a re-emerging cause of zoonotic tuberculosis in humans [3,4]. This disease poses a major threat to public health and creates socio-economic problems, including loss of meat due to seizures in slaughterhouses and lower milk yields [5,6]. Accordingly, bovine tuberculosis has an impact upon the international trade of animals and animal products [6,7], and bovine tuberculosis must be notified to the World Organization for Animal Health, as mentioned in its Terrestrial Animal Health Code [8].

In Algeria, some of the 2,049,652 heads of cattle [9] in the country are suspected to be infected by *M. bovis* [10]. Cases of bovine tuberculosis are often reported in slaughterhouse registers, and slaughterhouse workers exhibit a higher prevalence of *M. bovis* tuberculosis than the general population [11] (F. Tazerart, unpublished data). In Algeria, ante-mortem tuberculin skin test screening and post-mortem examinations for suspicious lesions at the slaughterhouse are the official diagnostic procedures for the disease [12]. Nevertheless, in the absence of systematic laboratory diagnosis, definitive confirmation of the disease is lacking. Consequently, bovine tuberculosis remains a neglected infectious disease in Algeria, and few microbiology studies have been reported [10]. The identification of the circulation of *M. bovis* strains and their hosts and geographical distribution are poorly understood.

This study aimed to start bridging this gap, using whole genome sequencing (WGS) to investigate a few *M. bovis* strains currently circulating in Algeria. 

## 2. Results

### 2.1. Data Analysis

Of the 928 cattle routinely slaughtered and inspected, 94 (10.13%, CI 8.24–12.34%) animals had visible lesions compatible with tuberculosis. The prevalence of suspected lesions significantly varied across the five studied slaughterhouses (*p* = 0.001). Prevalence was 20.9% at the Médéa slaughterhouse, which was the most affected site, and 5.8% at the Bgayet slaughterhouse, which was the least affected site (Figure 1). Males accounted for 617/928 (66.5%) of slaughtered animals and females for 311/928 (33.5%) (*p* < 10^−4^), although the prevalence of tuberculous lesions was higher in females than in males (*n* = 41/311 (13.18%) compared to n = 53/617 (8.59%), *p* = 0.038). Most of the slaughtered animals were aged <2 years (*n* = 427/928; 46%), followed by animals aged 2–5 years (*n* = 399/928; 43%) and animals older than 5 years (*n* = 102/928; 11%). Older animals >5 years were most affected (25.5%), followed by young animals (<2 years) at 9.83%, then adult animals (2–5 years) at 6.51%. The differences among prevalence values were significant (*p* < 10^−4^). Most slaughtered animals were of average weight (score 2.5–3) (536/928; 57.75%), followed by lean animals (286/928; 30.81%). The prevalence of the suspected lesions was 60/536 (11.2%) in average weight animals, 31/286 (10.8%) in lean animals and 03/106 (2.8%) in fatty animals (*p* = 0.03). Tuberculous lesions were mainly located on the lymph nodes (95.7%) and 4.3% in the viscera (Table 1). 

### 2.2. Genomic Typing Analysis

A total of 94 samples, including 90 lymph node samples, three lung samples and one liver sample, yielding 13 cultures obtained on Löwenstein–Jensen culture medium, Middlebrook 7H10 solid medium and Coletsos medium were identified as *M. tuberculosis* complex by partial *rpoB* PCR-sequencing. No other mycobacterial species was cultured from these samples. Further WGS refined the identification of the 13 isolates as *M. bovis*. Previously, the work of Loiseau et al. indicated that the diversity of *M. bovis* in Africa has been underestimated, after their discovery of new clonal complexes in the continent [13]. Genomic comparison of the Algerian *M. bovis* isolates to the ones reviewed in the work of Loiseau et al. indicated four different clonal complexes of *M. bovis* in Algeria—three of which were the newly discovered ones, and one of which was known—the Eu2 clonal complex commonly found in western Europe. Specifically, we found six isolates, namely Q1128, Q1131, Q1135, Q1138, Q1139 and Q1142, which were grouped together with reference clinical strains of *M. bovis* clonal complex-Unknown2. One isolate, Q1134, was related to *M. bovis* clonal complex-Unknown7; one isolate, Q1140, was related to *M. bovis* clonal complex-Unknown4; three isolates, Q1141, Q1129 and Q1136, belonged to *M. bovis* clonal complex-Europe 2; and there was one new clone for two isolates, Q1132 and Q1133, which did not belong to any of the clonal complexes identified by Loiseau et al. (Figure 2, Appendix A) [13]. Based on available data for 3364 clinical isolates of *M. bovis* (Appendix A) [13], African countries other than Algeria included Ethiopia, Malawi, Eritrea, Morocco, Tunisia and South Africa. European countries included the UK, Switzerland, Sweden, Spain, France, Italy, Germany, Netherlands and Belgium. American countries included the USA, Canada, Brazil and Mexico. Asian countries include Lebanon. The proportions of *M. bovis* genotypes observed in Algeria were significantly different than those observed in other countries (*p* < 0.001), other African countries (*p* = 0.045) and American countries (*p* < 0.001) but did not significantly differ from European countries (*p* = 0.245). Using a 12 SNPs cut-off to detect potential cross-transmission [14], we detected one group of transmission involving isolates Q1132 and Q1133 exhibiting 215-SNPs, and 0-SNPs after removing repetitive regions (Figure 2, Appendix A).

Further comparison of bovine isolates with four available Algerian patients’ *M. bovis* isolates previously determined as belonging to the European 2 group (P9978, P9979, P9980 and P9981) [15] indicated no evidence of zoonotic infection of these isolates alone using the same SNP cut-off as above (12-SNPs). Statistical information on the genome sequences are reported in Appendix A. Reads were deposed in GenBank with accession number PRJNA715078.

## 3. Discussion

In Algerian slaughterhouses, the inspection of slaughtered animals to look for lesions compatible with tuberculosis differs from one slaughterhouse to another.

Here, more males than females were slaughtered, in accordance with regulations prohibiting the slaughter of female animals other than in cases of necessity. However, we observed a significantly higher prevalence of tuberculosis in females than in males, an observation that may also reflect an increased susceptibility of females that go through gestation, parturition and lactation, and live a long, productive life [16]. Most slaughtered animals were under two years old, reflecting the high organoleptic value of meat in young animals, which is in great demand. However, older animals over the age of five years had a significantly higher prevalence of lesions suspected of being tuberculosis. Indeed, the chances of exposure to tuberculosis increases with age [17]. Most slaughtered animals were of average weight (score 2.5–3) reflecting consumer preference for meat containing a reasonable amount of fat. In addition, we observed a significant difference between the three body weight scores of animals suspected of tuberculosis; those of average weight were most affected, followed by those that were lean, then fatty animals, potentially explained by the wasting nature of this disease [18].

In this context, this report on the WGS-based identification and typing of a collection of *M. bovis* isolates updates the 2007 snapshot of bovine tuberculosis in Algeria [10]. The prevalence of bovine tuberculosis, as estimated from the veterinary inspection of slaughtered cattle, was 10.13%, higher than previously reported in Algeria in 2007 (3.6%) and in neighboring Morocco in 2014–2015 (3.7%) [10,19].

In this study, WGS revealed the presence of four different genotypes of *M. bovis* and one new genotype in Algeria compared to 12 genotypes reported by Loiseau et al. (Figure 2, Appendix A) [13]. We detected two isolates exhibiting a closely related WGS pattern, suggesting these two isolates, Q1132 and Q1133, were issued from the same *M. bovis* clone. These two isolates were isolated from the same slaughterhouse (Mouzaia in Blida).

Unsurprisingly, these data illustrate the cross-border spread of *M. bovis* and the fact that Algeria regularly imports live cattle from Europe, notably from France and Spain, each supplying 40,000 heads of cattle per year [20,21]. These data indicate that part of the epidemiology of bovine tuberculosis in cattle in Algeria is driven by the cross-border movement of animals. 

However, limited information about slaughtered animals, such as their origin and movements, limits the possibility of carrying out a spatial epidemiological study on the distribution of bovine tuberculosis. Illegal slaughter further complicates the establishment of such a bovine tuberculosis map. Additionally, only 13 *M. bovis* isolates were available for this first ever WGS analysis of animal *M. bovis* in Algeria. Such a small sample is likely to underestimate pathogen diversity in Algeria and is therefore unlikely to ascertain epidemiological linkages between cases. 

Finally, despite the above-reported limitations, this founding WGS-based study in Algeria indicates the contribution of WGS to tracing animal *M. bovis* isolates in that country. This study suggests that it would be of interest to establish an Algerian national *M. bovis* WGS database for tracing sources of infection, including the potential role of cross-border importation and of wild fauna, as reported in other countries such as the UK [22,23], and the cross-transmission in farms. Further comparisons between human [15] and animal isolates would also help to trace sources of zoonotic *M. bovis* tuberculosis in patients from a medical prevention perspective.

Given such a small number of isolates, this analysis could be very skewed, and a larger number of studied isolates may provide similarities with other locations. This indicates that continued research in this area is required.

Despite this limitation, we did find similarities with European nations, which is consistent with cattle being imported into Algeria from Europe and the historical links between these regions

## 4. Materials and Methods

### 4.1. Study Area and Sample Collection

The study was conducted during different periods of 2017 and 2018 in five slaughterhouses in Algeria, namely the slaughterhouses in Bgayet (January–May 2018), Sétif (January–May 2018), Médéa (December 2016–April 2017), Mouzaia (March–May 2017) and the slaughterhouses in Boufarik, which are in the department of Blida (Figure 1) (March–May 2018). The distance between each slaughterhouse was estimated to be between 27 km (Mouzaia–Boufarik) and 315 km (Sétif–Mouzaia). In detail, these 13 isolates were obtained from three slaughterhouses, namely the Mouzaia slaughterhouse in Blida (Q1128, Q1129, Q1131, Q1132 and Q1133), the Médéa slaughterhouse (Q1134, Q1135 and Q1136) and the Boufarik slaughterhouse in Blida (Q1138, Q1139, Q1140, Q1141 and Q1142). The choice of these slaughterhouses was justified by their accessibility and their relative size in the region in terms of cattle slaughter. Cattle slaughtered in these slaughterhouses came from the neighboring regions or sometimes from remote regions. An information sheet was systematically established for each animal slaughtered, including sex, age, breed, body score and origin. During the study period, lesions of suspected tuberculosis were collected from slaughtered cattle and conserved for 11–15 months at −20 °C until further processing.

### 4.2. Tissue Preparation and Culture

Some of these samples were processed at the Institut Pasteur d’Algérie in Algiers, and the rest were processed in the biosafety level 3 laboratory (BSL3) at the Institut Hospitalier Universitaire, Méditerranée Infection in Marseille. At the Institut Pasteur d’Algérie, gross visible lesions were thawed, dissected and crushed using a sterile pestle and mortar. Crushed tissues were decontaminated by adding 4 mL of 4% NaOH [24] and washed with sterile distilled water. Pellets were inoculated in a Löwenstein–Jensen culture medium containing glycerol (three tubes) and were incubated at 37 °C for 12 weeks. At the Institut Hospitalier Universitaire, gross visible lesions were thawed, and aliquots were decontaminated into a triple volume of 1% chlorhexidine (chlorhexidine digluconate, Sigma-Aldrich, St. Louis, MO, USA) [25] incubated for 30 min at room temperature with manual agitation every five minutes. After eliminating chlorhexidine, 10 mL of a neutralizing solution (1000 mL sterile phosphate buffered saline (PBS), 3 g egg lecithin and 10 mL Tween 80) were added, and the mixture was left at room temperature for 10 min with agitation to inactivate the remaining chlorhexidine. After three sterile PBS washes, the decontaminated tissue was finely crushed using Potter’s crusher and inoculated in four types of culture media including Middlebrook 7H10 solid medium containing 10% (*v*/*v*) oleic acid, albumin, dextrose and catalase (OADC) (Becton Dickinson, Sparks, MD, USA) and 0.5% (*v*/*v*) glycerol (Euromedex, Souffelweyersheim, France); 10% OADC-Middlebrook 7H10 solid medium containing sodium pyruvate (4.16 mg/mL) (Sigma-Aldrich) [26,27]; 10% OADC-Middlebrook 7H9 liquid medium (Becton Dickinson), containing 0.05% Tween 80 (Sigma-Aldrich) [28]; and Coletsos medium (Bio-Rad, Marnes-la-Coquette, France). Cultures were incubated at 37 °C for 12 weeks with a 10-day follow-up of colony growth.

### 4.3. Isolate Identification

Any culture growing on any of the solid media was dissolved in 200 μL PBS and inactivated at 100 °C for one hour before further handling in the biosafety level 2 (BSL2) laboratory. Total DNA was extracted by vortexing the suspension with glass powder (Sigma-Aldrich) using the FastPrep apparatus (MP Biomedicals, Santa Ana, CA, USA) and extracting DNA using a Qiagen kit (Qiagen, Courtaboeuf, France), as previously described [29]. Partial *rpoB* gene PCR-sequencing was then performed to confirm the *M. tuberculosis* complex, as previously described [30].

### 4.4. Genome Sequence Analyses

As for whole genome sequence (WGS) experiments, DNA was extracted using the InstaGene™ Matrix (Bio-Rad, Marnes-la-Coquette, France) and quantified using a Qubit assay with the high sensitivity kit (Life technologies, Carlsbad, CA, USA). A total of 0.2 µg/µL of DNA was then sequenced by Illumina MiSeq runs (Illumina Inc., San Diego, USA). DNA was fragmented and amplified by limited PCR (12 cycles), introducing dual-index barcodes and sequencing adapters. After purification on AMPure XP beads (Beckman Coulter Inc., Fullerton, USA), the libraries were normalized and pooled for sequencing on the MiSeq. Paired-end sequencing and automated cluster generation with dual indexed 2 × 250-bp reads were performed. Output sequencing reads were analyzed using MTBseq [31] to identify species, lineages and sub-lineages and to calculate the SNP distance between study strains. MTBseq was used also to extract and align the SNPs between the genome studies. Identification results were supported by other tools such as TB Profiler and Mykrobe Predictor-TB [32,33]. A phylogenetic tree based on extracted SNPs between 105 *M. bovis* genomes was generated using PhyML 3.0 online tools (http://phylogeny.lirmm.fr/phylo_cgi/one_task.cgi?task_type=phyml; accessed on 1 October 2020).

### 4.5. Statistical Analysis

The data collected were analyzed using SPSS (Statistical Package for Social Sciences, version 23.0) and Microsoft Office Excel software programs. As a criterion, reliability difference indicators used the profile *p* < 0.05. Statistical analyses were performed using Chi-square tests to compare the presence of risk factors between suspected cases of tuberculosis (Table 1) and to compare the proportions of *M. bovis* genotypes, among other genotypes in different geographic regions/countries. We delineated four groups of data: Group 1, all countries in the 3364-isolates database; Group 2, all African countries from the database; Group 3, all American countries; and Group 4, all European countries.

## Figures and Tables

**Figure 1 pathogens-10-00802-f001:**
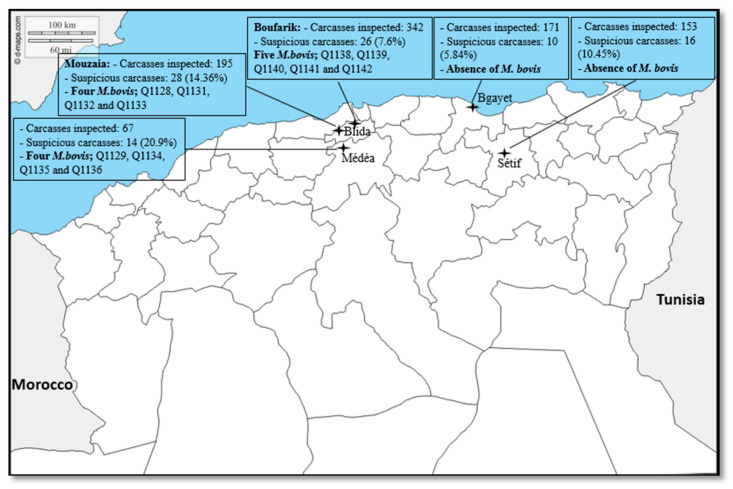
Location of the five slaughterhouses in four departments in northern Algeria.

**Figure 2 pathogens-10-00802-f002:**
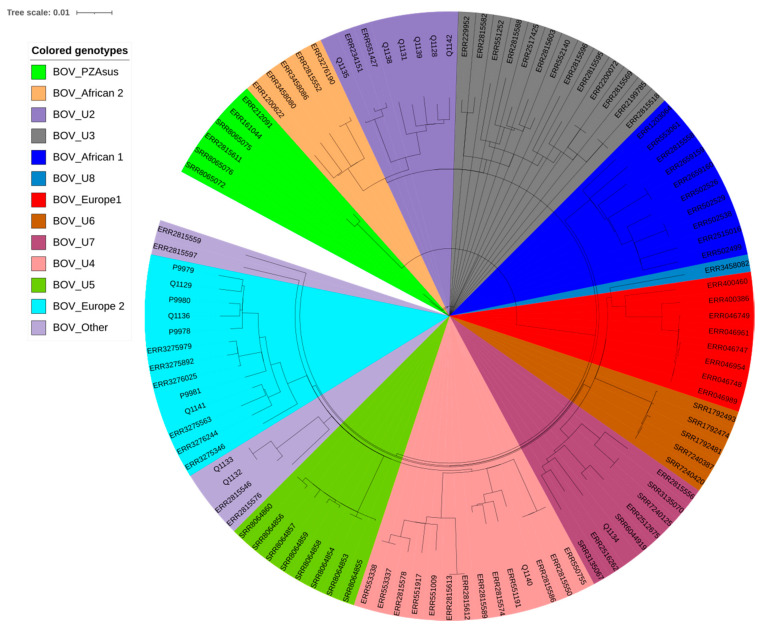
Phylogeny tree of 105 genome sequence of *Mycobacterium bovis* based on 2881 variable positions extracted using MTBseq. The scale bar indicates the number of substitutions per polymorphic site. The tree was generated using PhyML 3.0 online tools.

**Table 1 pathogens-10-00802-t001:** Distribution of suspected cases of tuberculosis according to the variation factors.

Variation Factors		Number of Suspect Carcasses	Rate (%)	Total	*p*
Gender	Male	53	8.59	617	0.038
	Female	41	13.18	311	
Age	Young (<2 years)	42	9.83	427	<10^−4^
	Adults (2–5 years)	26	6.51	399	
	Elderly (>5 years)	26	25.5	102	
Weight (score)	Lean (1–2)	31	10.83	286	0.03
	Middle (2.5–3)	60	11.19	536	
	Fat (3.5–5)	3	2.83	106	
Location	Lymph nodes	90	95.74	-	-
	Viscera	4 (3 lungs, 1 liver)	4.26	-	
Total		94	10.13	928	

## Data Availability

Not applicable.

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
