# Peer review of "Whole Genome Sequence Analysis of Mycobacterium bovis Cattle Isolates, Algeria"

_pathogens, 2021, doi:10.3390/pathogens10070802_

Round 1

Reviewer 1 Report

General Comment:
A selection of 13 strains of M. bovis isolated from cattle from different regions of Algeria was genome-sequenced and analysis of sequences was performed using the MTBseq pipeline. The results presented provide information on the clonal complex groupes of these M. bovis isolates.

It is a well-structured, well-documented and well-written study.

Minor Points:

  • In figure 2 the information regarding the groups Af2 and Eu2 could be indicated. Except for the Q1129 strain, the clustering presented in Fig. 2 seems to separate the two groups Af2 and Eu2. How to explain this result for the Q1129 strain? The authors should provide the detail of characteristic of the clonal complexes, the spoligotype signature and the phylogenetical marker for each strain.
  • I missed a detailed sequencing statistics report.
  • The ISMapper tool could be used (on the reads) to search for the abundance of IS6110 copies on this genomes panel combining animal and human strains in order to verify (or not) the correlation between copy number and adaptation to the host.

Author Response

Reviewer 1: 

Minor Points:

  • In figure 2 the information regarding the groups Af2 and Eu2 could be indicated. Except for the Q1129 strain, the clustering presented in Fig. 2 seems to separate the two groups Af2 and Eu2. How to explain this result for the Q1129 strain? The authors should provide the detail of characteristic of the clonal complexes, the spoligotype signature and the phylogenetical marker for each strain.
  • I missed a detailed sequencing statistics report.
  • The ISMapper tool could be used (on the reads) to search for the abundance of IS6110 copies on this genomes panel combining animal and human strains in order to verify (or not) the correlation between copy number and adaptation to the host.

Author’s Response: The authors fully agree with the reviewer’s comment that these above-mentioned identification methods are not optimal ones to discriminate the different announced genotypes of M. bovis. Therefore, we compared the genome sequences of M. bovis isolates under study with 88 clinical reference isolates belonged to 12 genotypes of M. bovis, as announced by Loiseau C. et al., 2020 (Reference 13).

According to this remark, the authors corrected the sentence to weight our point as follows: “Further WGS refined the identification of the 13 isolates as M. bovis. In details, comparative genome analysis with 88 references genomes of M. bovis belonged to 12 announced genotypes (Table S1) (13), found six isolates Q1128, Q1131, Q1135, Q1138, Q1139, and Q1142 grouped together with reference clinical strains of M. bovis genotype-Unknown2, one isolate Q1134 was related to M. bovis genotype-Unknown7, one isolate Q1140 was related to M. bovis genotype-Unknown4, three isolates Q1141, Q1129 and Q1136 belonged to M. bovis genotype-Europe 2 and two non-typeable isolates Q1132 and Q1133 were grouped together (Figure 2, Table S1).” (Lines 80-87).

More, we replaced the Figure 2 “heatmap clustering” by a new Figure “phylogenetic tree of 105 genomes sequences of Mycobacterium bovis board on 2,881 variable positions extracted using MTBseq” (Lines 91-93). In addition, SNPs distance between 105 M. bovis were added as Table S2 in Supplementary data. Following these new results, the discussion and abstract have been amended.

Reviewer 2 Report

Dear Authors,

Thank you for your manuscript 'Whole genome analysis of Mycobacterium bovis cattle isolates, Algeria.' See below for my thoughts on your study.

  1. In the Abstract, I would change the following sentences to improve clarity?

'Two isolates, both from Blida, exhibited no single nucleotide polymorphism...'  I think you should say both isolates exhibited no pairwise SNP differences instead. The isolates themselves do have SNPs, so the original sentence is misleading.

'None of these 13 isolates showed similarity with four zoonotic M. bovis...'  I would change this to 'None of these 13 isolates were closely related to four zoonotic M. bovis isolates'.

  1. In the Introduction, line 48, please change 'sequence' to 'sequencing'.
  2. Results: Some more detail on your samples is required here I feel. You had access to data from 928 slaughtered animals from 5 abattoirs. 94 exhibited tuberculous like pathology, but only 13 were confirmed by culture? This seems quite a small number - in the UK we would see ~40% of lesioned animals presenting with positive M. bovis culture. I may be reading this incorrectly however - these 13 were confirmed M. bovis?  Did you have any other culture positive animals which were confirmed to be other tuberculous mycobacteria? If so, it might be good to detail what they were before then focusing on just the M. bovis isolates.
  3. Results, Table 1: You have done some statistical analyses using chi squared tests and Fishers test on these differences in proportions in M. bovis prevalence, but there are no indications of significance of tests appended to the table. It would aid understanding of the manuscript if you did add these in.
  4. Results Section, Genome Typing Analysis Section 2.2., Line 84: You mention using a 12 SNP cut off to detect potential evidence of 'cross transmission'.  I'd perhaps change that phrase to 'epidemiological association'. In addition, I think perhaps you should discuss the pairwise SNP difference thresholds needed to make epidemiological associations.  A 12 SNP difference seems a large threshold to infer recent transmission given how slowly these bacteria mutate.  See Meehan et al EBiomedicine Volume 37, November 2018, Pages 410-416 for some useful discussion of this.  I think a smaller threshold would be better.  Given that you don't observe any very close genetic relationships between most of the 13 isolates you discuss, reducing your threshold isn't going to radically change your interpretation of your data.  And even for the 2 isolates which are closely related, the pairwise SNP distance is zero so well below a threshold of 12, 5 or even 1 SNP.
  5. I very much like your illustration of the pairwise SNP distance matrix in Figure 2.  It's very easy to read.  I think having a more traditional phylogeny included alongside it would be helpful though.  In addition, some indication of the abattoir location from which each isolate comes would be helpful here.  Do you find that isolates from the same abattoir cluster together on your tree in distinct clades? That would be an interesting finding considering M. bovis has been observed to cluster geographically in other locations - with closely related isolates tending to have home ranges.  I know you say in the methods section that abattoirs do process animals from further away than their location, so that may be an issue here. But if you did see that more closely related isolates on the tree were from the same abattoir it might be a suggestion that there is some geographical clustering occurring. Alternatively, if you see isolates from multiple, dispersed abattoirs appearing in the same clades, then that may confirm that abattoir surveillance is unable to detect clustering of related types. Either way, I think in your conclusion, it is worth saying that this is an initial step in setting up genomic surveillance, a proof of concept.  But to get the most from these data for epidemiological surveillance, accessing geo-locations for the origins of animals would be crucial.     
  6. Results, Line 92: It's perhaps premature to say there is no epidemic of zoonotic infection on the strength of the isolates presented.  Its more likely that your small sampling has not captured the full diversity of the pathogen thereby preventing you finding firm evidence of potential links between human and animal cases.  Again, in your conclusions, I think making this clear would be useful.  What you have done is a necessary first step and proof of concept.  It now needs to establish a more systematic sampling approach to be useful from a disease surveillance and human health perspective.
  7. Discussion Section: As with my points above, I think you need to talk more about the limitations of the study as presented, but also talk about how it is a useful proof of concept, that with more systematic sampling and geographical origin data, could greatly aid disease surveillance, control and public health.

You don't mention the two closely related isolates Q1132 and Q1133 in the discussion.  I think the fact they exhibit such similarity compared to the rest of your dataset merits some discussion.  Which abattoir were they collected at?  It could that be evidence of a recent transmission between two cattle and if they come from roughly the same location, that is noteworthy and further proof of the usefulness of what you have established.  

In line 135, you mention that two of your isolates are from the Western European EU2 lineage. You the go on to mention contemporary cross border trade with France, Spain and Brazil which all have extant EU2 lineage M. bovis within their borders.  So contemporary trade for recent introduction is possible.  Given Algeria's history with France, is there also the possibility of import of animals with EU2 bacteria in the past? Such introductions may have established foci of infection in Algeria? We know that similar introductions from the colonial past introduced EU2 lineages into south America (see Zimpel et al Front. Microbiol., 07 May 2020 | https://doi.org/10.3389/fmicb.2020.00843) - so perhaps something similar is possible for Algeria?

Again, you could use this as a way to discuss future improvements to the programme of genome sequencing you propose.  Having a good enough database in place of M. bovis diversity in Algeria could help you to identify endemic lineages which have been present for some time in the country and then help to identify more recent introductions brought in by trade.  Being able to distinguish between these two things would be crucial for any disease surveillance scheme that seeks to find the source for local outbreaks of infection.

  1. Methods Section: Some more detail on the bioinformatic approaches used here would be useful.  Which alignment and SNP calling functions were used?  Which repetitive regions were excluded?  The PE/PPE regions?  Indications of sequence quality, read depth an coverage. 

Author Response

Reviewer 2: 1. In the Abstract, I would change the following sentences to improve clarity?

'Two isolates, both from Blida, exhibited no single nucleotide polymorphism...'  I think you should say both isolates exhibited no pairwise SNP differences instead. The isolates themselves do have SNPs, so the original sentence is misleading.

Author's Response: Corrected accordingly. (Line 23)

Reviewer 2: 'None of these 13 isolates showed similarity with four zoonotic M. bovis...'  I would change this to 'None of these 13 isolates were closely related to four zoonotic M. bovis isolates'.

Author's Response: Corrected accordingly. (Line 24)

Reviewer 2: 2.  In the Introduction, line 48, please change 'sequence' to 'sequencing'.

Author's Response: Corrected accordingly. (Line 48)

Reviewer 2: 3.  Results: Some more detail on your samples is required here I feel. You had access to data from 928 slaughtered animals from 5 abattoirs. 94 exhibited tuberculous like pathology, but only 13 were confirmed by culture? This seems quite a small number - in the UK we would see ~40% of lesioned animals presenting with positive M. bovis culture. I may be reading this incorrectly however - these 13 were confirmed M. bovis?  Did you have any other culture positive animals which were confirmed to be other tuberculous mycobacteria? If so, it might be good to detail what they were before then focusing on just the M. bovis isolates.

Author’s Response: The authors do confirm the statistics that of 928 animals slaughtered in 5 slaughterhouses, 94 showed suspicious tuberculosis lesions. The authors clarify that only 94 samples including 90 lymph node, 03 lung and 01 liver samples (Line 76) were cultured and 13 gave a positive culture confirmed as M. bovis by WGS (Lines 80). The authors now clarify that no other mycobacterium other than M. bovis has been detected (Line 79).

Reviewer 2: 4. Results, Table 1: You have done some statistical analyses using chi squared tests and Fishers test on these differences in proportions in M. bovis prevalence, but there are no indications of significance of tests appended to the table. It would aid understanding of the manuscript if you did add these in.

Author’s Response: Corrected accordingly (Table 1).

Reviewer 2: 5. Results Section, Genome Typing Analysis Section 2.2., Line 84: You mention using a 12 SNP cut off to detect potential evidence of 'cross transmission'.  I'd perhaps change that phrase to 'epidemiological association'. In addition, I think perhaps you should discuss the pairwise SNP difference thresholds needed to make epidemiological associations.  A 12 SNP difference seems a large threshold to infer recent transmission given how slowly these bacteria mutate.  See Meehan et al EBiomedicine Volume 37, November 2018, Pages 410-416 for some useful discussion of this.  I think a smaller threshold would be better.  Given that you don't observe any very close genetic relationships between most of the 13 isolates you discuss, reducing your threshold isn't going to radically change your interpretation of your data.  And even for the 2 isolates which are closely related, the pairwise SNP distance is zero so well below a threshold of 12, 5 or even 1 SNP.

Author’s Response: Following this remark, the authors compared studied isolates at genomic level with 88 clinical reference isolates belonged to 12 genotypes of M. bovis announced by Loiseau C. et al., 2020 (Reference 13). Accordingly, we replaced the Figure 2 “heatmap clustering” by a new Figure “phylogenetic tree of 105 genomes sequences of Mycobacterium bovis board on 2,881 variable positions extracted using MTBseq” (Lines 94-97). In addition, SNPs distance between 105 M. bovis were added as Table S2 in Supplementary data. Following these new results, the discussion and abstract have been amended.

More, we clarify the isolation source of isolates: “In details, these 13 isolates were obtained from three slaughterhouses, Mouzaia slaughterhouse in Blida (Q1128, Q1129, Q1131, Q1132, and Q1133), Médéa slaughterhouse (Q1134, Q1135 and Q1136) and Boufarik slaughterhouse in Blida (Q1138, Q1139, Q1140, Q1141 and Q1142)”. (lines 169-172).

Reviewer 2: 7. Results, Line 92: It's perhaps premature to say there is no epidemic of zoonotic infection on the strength of the isolates presented.  Its more likely that your small sampling has not captured the full diversity of the pathogen thereby preventing you finding firm evidence of potential links between human and animal cases.  Again, in your conclusions, I think making this clear would be useful.  What you have done is a necessary first step and proof of concept.  It now needs to establish a more systematic sampling approach to be useful from a disease surveillance and human health perspective.

Author’s Response: The reviewer is perfectly right, and this paragraph was modified accordingly (Line 98).

Reviewer 2: 8. Discussion Section: As with my points above, I think you need to talk more about the limitations of the study as presented, but also talk about how it is a useful proof of concept, that with more systematic sampling and geographical origin data, could greatly aid disease surveillance, control and public health.

Author’s Response: The reviewer is perfectly right and both limitations and perspectives are further discussed (Lines 157-164). Despite the above-reported limitations, this founding WGS-based study in Algeria illustrated the contribution of WGS tracing animal M. bovis isolates in that country. This study suggested it would be of interest to establish an Algerian national M. bovis WGS database for tracing sources of infection including the potential role of cross-border importation and wild fauna as reported in other countries such as UK [22,23]; and the cross-transmission in farms. Also, further comparisons between human [15] and animal isolates would help trace sources of zoonotic M. bovis tuberculosis in patients, from a medical prevention perspective.

Reviewer 2: You don't mention the two closely related isolates Q1132 and Q1133 in the discussion.  I think the fact they exhibit such similarity compared to the rest of your dataset merits some discussion.  Which abattoir were they collected at?  It could that be evidence of a recent transmission between two cattle and if they come from roughly the same location, that is noteworthy and further proof of the usefulness of what you have established.  

Author’s Response: We detected two isolates exhibiting a closely related WGS pattern, suggesting these two isolates Q1132 and Q1133 were issued from the same M. bovis clone. These two isolates were isolated from the same slaughterhouse (Mouzaia of Blida), but we do not know if animals issued from the same breeding. This point is now clarified in Lines 140-143.

Reviewer 2: In line 135, you mention that two of your isolates are from the Western European EU2 lineage. You the go on to mention contemporary cross border trade with France, Spain and Brazil which all have extant EU2 lineage M. bovis within their borders.  So contemporary trade for recent introduction is possible.  Given Algeria's history with France, is there also the possibility of import of animals with EU2 bacteria in the past? Such introductions may have established foci of infection in Algeria? We know that similar introductions from the colonial past introduced EU2 lineages into south America (see Zimpel et al Front. Microbiol., 07 May 2020 | https://doi.org/10.3389/fmicb.2020.00843) - so perhaps something similar is possible for Algeria?

Again, you could use this as a way to discuss future improvements to the programme of genome sequencing you propose.  Having a good enough database in place of M. bovis diversity in Algeria could help you to identify endemic lineages which have been present for some time in the country and then help to identify more recent introductions brought in by trade.  Being able to distinguish between these two things would be crucial for any disease surveillance scheme that seeks to find the source for local outbreaks of infection.

Author’s Response: Following this remark, the authors compared studied isolates at genomic level with 88 clinical reference isolates belonged to 12 genotypes of M. bovis announced by Loiseau C. et al., 2020 (Reference 13).

Accordingly, we have modified our sentence to weight our point as follows: “Further WGS refined the identification of the 13 isolates as M. bovis. In details, comparative genome analysis with 88 references genomes of M. bovis belonged to 12 announced genotypes (Table S1) (13), found six isolates Q1128, Q1131, Q1135, Q1138, Q1139, and Q1142 grouped together with reference clinical strains of M. bovis genotype-Unknown2, one isolate Q1134 was related to M. bovis genotype-Unknown7, one isolate Q1140 was related to M. bovis genotype-Unknown4, three isolates Q1141, Q1129 and Q1136 belonged to M. bovis genotype-Europe 2 and two non-typeable isolates Q1132 and Q1133 were grouped together (Figure 2, Table S1).” (Lines 80-87).

More, we replaced the Figure 2 “heatmap clustering” by a new Figure “phylogenetic tree of 105 genomes sequences of Mycobacterium bovis board on 2,881 variable positions extracted using MTBseq” (Lines 91-94). In addition, SNPs distance between 105 M. bovis were added as Table S2 in Supplementary data. Following these new results, the discussion and abstract have been amended.

As comparative results between common detected genotypes, we found that Algeria had a different distribution than the other countries (p< 0.001), than other African countries (p= 0.045) and American countries (p<0.001); but it had the same distribution (p=0.245) than Europe countries (Lines 142-144).

Reviewer 2: 9. Methods Section: Some more detail on the bioinformatic approaches used here would be useful.  Which alignment and SNP calling functions were used?  Which repetitive regions were excluded?  The PE/PPE regions?  Indications of sequence quality, read depth and coverage. 

Authors’ answer: In this study we used “MTBseq tools” in order to extract and alignment the SNPs between genomes (Line 219). The excluded repetitive regions were cited in the MTBseq article (Reference 31). More, we added all statistical data including quality, coverage…. in the Table S1 in Supplementary data.     

Reviewer 3 Report

The manuscript concerns bovine tuberculosis, an important disease in cattle. Although the study is local in nature, it shows that information on bovine tuberculosis in Algeria is very limited. A clear limitation of the study is the characteristics of only 13 Mycobacterium bovis strains, although a significant number of cattle were tested. However, due to a well-conducted analysis of these strains, the manuscript can be published. In the Materials and Methods section, there is no information on the number of animals from which samples were collected for further analysis, so the authors should complete this data. I do not have any substantial comments to the study.

Author Response

Reviewer 3: The manuscript concerns bovine tuberculosis, an important disease in cattle. Although the study is local in nature, it shows that information on bovine tuberculosis in Algeria is very limited. A clear limitation of the study is the characteristics of only 13 Mycobacterium bovis strains, although a significant number of cattle were tested. However, due to a well-conducted analysis of these strains, the manuscript can be published. In the Materials and Methods section, there is no information on the number of animals from which samples were collected for further analysis, so the authors should complete this data. I do not have any substantial comments to the study.

Author’s Response: The reviewer is perfectly right:

  • The authors now clarify that 94 samples including 90 lymph node, 03 lung and 01 liver samples (Line 76) have been cultured, to yield 13 bovis isolates (Line 80).
  • The authors acknowledge the small number of isolates (13) in the limitations of the study (Lines 110-111).
  • The authors acknowledge this overall positive comment.

Round 2

Reviewer 2 Report

Dear Authors,

Thank you for your consideration of my suggested edits and the additions you have made to the manuscript.  I think your addition of the phylogenetic tree and comparison to other M. bovis isolates is useful.

For clarity I still think you need to make some alterations to the text.

As before in the Abstract, you say that "Two isolates from Blida, exhibited no single nucleotide polymorphism".  Please change this to 'exhibited no pairwise differences in single nucleotide polymorphism'.  It is just wrong to say both isolates exhibit no SNPs.

In Line 81, you discuss the comparison of Alergian M. bovis isolates to other M. bovis isolates from the work of Loiseau et al.  Loiseau et al found greater diversity in African M.bovis than had previously been documented - I think you should mention this and say that your phylogeny indicates some of the newly identified clonal complexes are present in your Algerian sampling.  You can make the point that African diversity has typically been underestimated and is another good reason to establish genomic surveillance in Algeria.

Consider changing Lines 80-87 to the following for clarity 'Previously, the work of Loiseau et al has indicated that the diversity of M. bovis in Africa has been underestimated, as per their discovery of new clonal complexes in the continent. Genomic comparison of Algerian isolates to M. bovis from the work of Loiseau et al indicated the presence of 4 different clonal complexes of M. bovis in Algeria - 3 of which were the newly discovered ones with one known - the Eu2 clonal complex commonly found in western Europe. Specifically we foundsix isolates Q1128, Q1131, Q1135, Q1138, Q1139, and Q1142 grouped together with reference clinical strains of M. bovis clonal complex-Unknown2, one isolate Q1134 was related to M. bovis clonal complex-Unknown7, one isolate Q1140 was related to M. bovis clonal complex-Unknown4, three isolates Q1141, Q1129 and Q1136 belonged to M. bovis clonal complex-Europe 2 and two non-typeable?? Isolates Q1132 and Q1133 were grouped together (Figure 2, Table S1).

What do you mean by non-typeable in the sentence above? You have genomic typing data?  Do you mean that these two isolates do not belong to any of the clonal complexes identified by Loiseau et al?  If so please state that is the case.

Also, you refer to these new clonal complexes as 'announced genotypes'.  I think referring to them as 'newly discovered clonal complexes' would be more appropriate.  

Line 97: Change study to studied

In Line 107, you add that the study is only based on 13 isolates.  As I suggested before, you need to add more detail here. I think you need to say that such a small sampling is an underestimate of the pathogen diversity in Algeria, and that it is therefore unlikely to find cases of epidemiological linkage between cases - similarly you cannot say there is no evidence of zoonotic infection on the strength of these isolates alone.

Lines 131-132: Please change 'announced genotypes' to 'newly discovered clonal complexes' as above.  

Line 137: I don't understand what you mean here.  Both of these isolates are genetically identical, both came from the same abattoir.  What do you mean by 'breeding'?  Do you mean both isolates may have come from the same animal, but you can't be sure?  I'm sorry I can't follow what is meant.

Line 138-144: I don't follow what you have done here, and I can't see a clear reference to the analysis you describe here in the methods section.  Are you comparing the distribution of SNP distances between Algerian isolates and other M. bovis isolates from around the world to find isolates which are more closely related?  You're wanting to demonstrate how more closely related your isolates are to European ones I assume judging by the p values you describe? It isn't clear what you've done here.

Also, these findings should be described in the results section, not the Discussion. 

 Author Response

Reviewer 2:

Dear Authors,

Thank you for your consideration of my suggested edits and the additions you have made to the manuscript.  I think your addition of the phylogenetic tree and comparison to other M. bovis isolates is useful.

For clarity I still think you need to make some alterations to the text.

As before in the Abstract, you say that "Two isolates from Blida, exhibited no single nucleotide polymorphism".  Please change this to 'exhibited no pairwise differences in single nucleotide polymorphism'.  It is just wrong to say both isolates exhibit no SNPs.

Author’s Response:  Corrected accordingly (Line 23)

In Line 81, you discuss the comparison of Alergian M. bovis isolates to other M. bovis isolates from the work of Loiseau et al.  Loiseau et al found greater diversity in African M.bovis than had previously been documented - I think you should mention this and say that your phylogeny indicates some of the newly identified clonal complexes are present in your Algerian sampling.  You can make the point that African diversity has typically been underestimated and is another good reason to establish genomic surveillance in Algeria.

Author’s Response: Following this remark, the authors changed the sentence (Lines 80-92) and add a new sentence (Lines 143-145).

Reviewer 2: Consider changing Lines 80-87 to the following for clarity 'Previously, the work of Loiseau et al has indicated that the diversity of M. bovis in Africa has been underestimated, as per their discovery of new clonal complexes in the continent. Genomic comparison of Algerian isolates to M. bovis from the work of Loiseau et al indicated the presence of 4 different clonal complexes of M. bovis in Algeria - 3 of which were the newly discovered ones with one known - the Eu2 clonal complex commonly found in western Europe. Specifically, we found six isolates Q1128, Q1131, Q1135, Q1138, Q1139, and Q1142 grouped together with reference clinical strains of M. bovis clonal complex-Unknown2, one isolate Q1134 was related to M. bovis clonal complex-Unknown7, one isolate Q1140 was related to M. bovis clonal complex-Unknown4, three isolates Q1141, Q1129 and Q1136 belonged to M. bovis clonal complex-Europe 2 and two non-typeable?? Isolates Q1132 and Q1133 were grouped together (Figure 2, Table S1).

What do you mean by non-typeable in the sentence above. You have genomic typing data?  Do you mean that these two isolates do not belong to any of the clonal complexes identified by Loiseau et al?  If so please state that is the case.

Author’s Response: Following this remark, the authors changed the sentence (Lines 80-92).

Reviewer 2: Also, you refer to these new clonal complexes as 'announced genotypes'.  I think referring to them as 'newly discovered clonal complexes' would be more appropriate. 

Author’s Response: Corrected accordingly (Lines 147-148).

Reviewer 2: Line 97: Change study to studied

Author’s Response: Corrected accordingly (Line 108).

Reviewer 2: In Line 107, you add that the study is only based on 13 isolates.  As I suggested before, you need to add more detail here. I think you need to say that such a small sampling is an underestimate of the pathogen diversity in Algeria, and that it is therefore unlikely to find cases of epidemiological linkage between cases - similarly you cannot say there is no evidence of zoonotic infection on the strength of these isolates alone. 

Author’s Response: Corrected accordingly (Line 108 and Lines 119-121).

Reviewer 2: Lines 131-132: Please change 'announced genotypes' to 'newly discovered clonal complexes' as above.  

Author’s Response: Corrected accordingly (Lines 147-148).

Reviewer 2: Line 137: I don't understand what you mean here.  Both of these isolates are genetically identical, both came from the same abattoir.  What do you mean by 'breeding'?  Do you mean both isolates may have come from the same animal, but you can't be sure?  I'm sorry I can't follow what is meant.

Author’s Response: These two isolates (Q1132 and Q1133) were isolated from the same slaughterhouse (Mouzaia of Blida), but we do not know whether the animals came from same breeding, or these were two isolates that came from the same animal.

Reviewer 2: Line 138-144: I don't follow what you have done here, and I can't see a clear reference to the analysis you describe here in the methods section.  Are you comparing the distribution of SNP distances between Algerian isolates and other M. bovis isolates from around the world to find isolates which are more closely related?  You're wanting to demonstrate how more closely related your isolates are to European ones I assume judging by the p values you describe? It isn't clear what you've done here.

Also, these findings should be described in the results section, not the Discussion. 

Author’s Response: Based on published data (Genotypes and isolation countries) we used the chi2 test to compare the proportion of common M. bovis genotypes among other genotypes between Algeria (this study) and other continents (Table S3). Also, these findings have been moved to the Result section (L92-L99).

Round 3

Reviewer 2 Report

Dear Authors, 

Thank you for attending to my previous comments and for your replies, it is much clearer now what you mean in your manuscript.  I have a few more minor issues I need to ask you to address:

1 - In your Methods Section, you mention the use of Fishers exact tests and chi squared tests, I think you need to explicitly say what you analysed with these tests as it is not obvious form the text. Specifically, you performed these analyses on (i) the animal level analyses of risk factors detailed in Table 1 and also (ii) the comparison of the proportions of different genotypes in different geographic regions / countries.  Please insert two short sentences in the methods statistical analysis section that simply say what you compared.  Be specific about which groups of strains were compared in your analysis.  From your text I am assuming you have assembled 4 groups of strain data - Group 1 all countries in the 3364 isolate database you quote. Group 2 all African countries from the database.  Group 3 All American countries and Group 4 All European.  This needs to be described in the methods section.  Also, what 'genotype' proportions are you comparing? Is it the clonal complex definition?  If so, please say.  If a different type of genetic information, please say in the methods what you are comparing as it is not obvious.

2 - Line 97: Please change the sentence to 'The proportions of M. bovis genotypes observed in Algeria is significantly different than that observed in other countries’.

3 - Line 99: Change 'different' to 'differ' and 'Europe' to 'European'.

4 - Given that you only have 13 Algerian isolates to compare, I think you need to be careful in interpreting the analysis of similarity to other geographical regions.  With such a small number of isolates the analysis could be very skewed, and with a larger number you may see similarities with other places.  I think you should say this and also say that this would be a good reason to do more work in this area.  Your finding of links to Europe is still interesting though, so you could say something like 'despite this limitation, we still find similarities with European nations, which is consistent with cattle import into Algeria from Europe and historical links between these regions'.

Thank you

Author Response

Reviewer 2: 1 - In your Methods Section, you mention the use of Fishers exact tests and chi squared tests, I think you need to explicitly say what you analysed with these tests as it is not obvious form the text. Specifically, you performed these analyses on (i) the animal level analyses of risk factors detailed in Table 1 and also (ii) the comparison of the proportions of different genotypes in different geographic regions / countries.  Please insert two short sentences in the methods statistical analysis section that simply say what you compared.  Be specific about which groups of strains were compared in your analysis.  From your text I am assuming you have assembled 4 groups of strain data - Group 1 all countries in the 3364 isolate database you quote. Group 2 all African countries from the database.  Group 3 All American countries and Group 4 All European.  This needs to be described in the methods section.  Also, what 'genotype' proportions are you comparing? Is it the clonal complex definition?  If so, please say.  If a different type of genetic information, please say in the methods what you are comparing as it is not obvious.

Author’s Response: The reviewer is perfectly right, and these methodological points have been clarified, correcting sentence in lines 242-243 and adding a new sentence (Lines 243-247). We compared the proportion of the M. bovis genotypes among the other genotypes between the different groups (Lines 244-245).

Reviewer 2: 2 - Line 97: Please change the sentence to 'The proportions of M. bovis genotypes observed in Algeria is significantly different than that observed in other countries’.

Author’s Response: Corrected accordingly (L96-98).

Reviewer 2: 3 - Line 99: Change 'different' to 'differ' and 'Europe' to 'European'.

Author’s Response: Corrected accordingly (L99).

Reviewer 2: 4 - Given that you only have 13 Algerian isolates to compare, I think you need to be careful in interpreting the analysis of similarity to other geographical regions.  With such a small number of isolates the analysis could be very skewed, and with a larger number you may see similarities with other places.  I think you should say this and also say that this would be a good reason to do more work in this area.  Your finding of links to Europe is still interesting though, so you could say something like 'despite this limitation, we still find similarities with European nations, which is consistent with cattle import into Algeria from Europe and historical links between these regions'.

Author’s Response: The reviewer is again perfectly right, the authors clarified this point in lines 168-170 and lines 172-174.